# All Word Embeddings from One Embedding

**Sho Takase**
Tokyo Institute of Technology
sho.takase@nlp.c.titech.ac.jp

**Sosuke Kobayashi**
Tohoku University    Preferred Networks, Inc.
sosk@preferred.jp

## Abstract

In neural network-based models for natural language processing (NLP), the largest part of the parameters often consists of word embeddings. Conventional models prepare a large embedding matrix whose size depends on the vocabulary size. Therefore, storing these models in memory and disk storage is costly. In this study, to reduce the total number of parameters, the embeddings for all words are represented by transforming a shared embedding. The proposed method, **ALONE** (**al**l word embeddings from **one**), constructs the embedding of a word by modifying the shared embedding with a filter vector, which is word-specific but non-trainable. Then, we input the constructed embedding into a feed-forward neural network to increase its expressiveness. Naively, the filter vectors occupy the same memory size as the conventional embedding matrix, which depends on the vocabulary size. To solve this issue, we also introduce a memory-efficient filter construction approach. We indicate our ALONE can be used as word representation sufficiently through an experiment on the reconstruction of pre-trained word embeddings. In addition, we also conduct experiments on NLP application tasks: machine translation and summarization. We combined ALONE with the current state-of-the-art encoder-decoder model, the Transformer [36], and achieved comparable scores on WMT 2014 English-to-German translation and DUC 2004 very short summarization with less parameters[1].

## 1  Introduction

Word embeddings have played a crucial role in the recent progress in the area of natural language processing (NLP) [3, 17, 22, 23]. In particular, word embeddings are necessary to convert discrete input representations into vector representations in neural network-based NLP methods [3]. To convert an input word $w$ into a vector representation in a conventional way, we prepare a one-hot vector $v_w$ whose dimension size is equal to the vocabulary size $V$ and an embedding matrix $E$ whose shape is $D_e \times V$ ($D_e$ represents a word embedding size). Then, we multiply $E$ and $v_w$ to obtain a word embedding $e_w$.

NLP researchers have used word embeddings as input in their models for several applications such as language modeling [38], machine translation [30], and summarization [26]. However, in these methods, the embedding matrix forms the largest part of the total parameters, because $V$ is much larger than the dimension sizes of other weight matrices. For example, the embedding matrix makes up one-fourth of the parameters of the Transformer, a state-of-the-art neural encoder-decoder model, on WMT English-to-German translation [36]. Thus, if we can represent each word with fewer parameters without significant compromises on performance, we can reduce the model size to conserve memory space. If we save the memory space for embeddings, we can also train a bigger model to improve the performance.

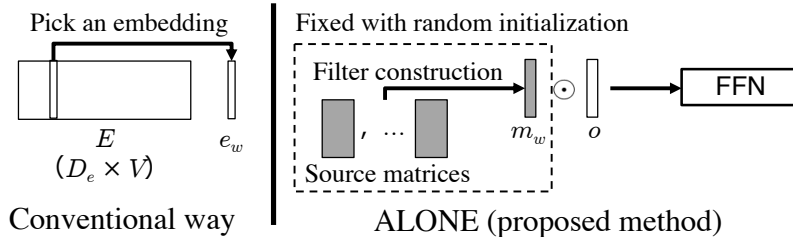

Figure 1: Word embedding constructions by the conventional way and our proposed ALONE. ALONE represents each word with an embedding $o$ and the filter vector $m_w$. We fix source matrices with random initialized values, and thus it is unnecessary to prepare trainable parameters for construction of $m_w$. To increase the expressiveness, we input the embedding into a feed-forward neural network.

To reduce the size of neural network models, some studies have proposed the pruning of unimportant connections from a trained network [15, 8, 39]. However, their approaches require twice or much computational cost, because we have to train the network before and after pruning. Another approach [31, 28, 1] limited the number of embeddings by utilizing the composition of shared embeddings. These methods represent an embedding by combining several primitive embeddings, whose number is less than the vocabulary. However, since they require learning the combination of assignments of embeddings to each word, they need additional parameters during the training phase. Thus, prior approaches require multiple training steps and/or additional parameters that are necessary only during the training phase.

To address the above issues, we propose a novel method: **ALONE** (**al**l word embeddings from **one**), which can be used as a word embedding set without multiple training steps and additional trainable parameters to assign each word to a unique vector. ALONE computes an embedding for each word (as a replacement for $e_w$) by transforming a shared base embedding with a word-specific filter vector and a shared feed-forward network. We represent the filter vectors with combinations of primitive random vectors, whose total is significantly smaller than the vocabulary size. In addition, it is unnecessary to train the assignments, because we assign the random vectors to each word randomly. Therefore, while ALONE retains its expressiveness, its total parameter size is much smaller than the conventional embedding size, which depends on the vocabulary size due to $D_e \times V$.

Through experiments, we demonstrate ALONE can be used for NLP with comparable performances to the conventional embeddings but fewer parameters. We first indicate ALONE has enough expressiveness to represent the existing word embedding (GloVe [22]) through a reconstruction experiment. In addition, on two NLP applications of machine translation and summarization, we demonstrate ALONE with the state-of-the-art encoder-decoder model trained in an end-to-end manner achieved comparable scores with fewer parameters than models with the conventional word embeddings.

## 2   Proposed Method: ALONE

Figure 1 shows an overview of the conventional and proposed word embedding construction approaches. As shown in this figure, the conventional way requires a large embedding matrix $E \in \mathbb{R}^{D_e \times V}$, where each column vector is assigned to a word, and we pick a word embedding with a one-hot vector $v_w \in \mathbb{R}^{1 \times V}$. In other words, each word has a word-specific vector, whose size is $D_e$, and the total size summed over the vocabulary is $D_e \times V$. To reduce the size, ALONE make words share some type of vector element with each other. ALONE has a base embedding $o \in \mathbb{R}^{1 \times D_o}$, which is shared by all words, and represents each word by transforming the base embedding. To concretely obtain a word representation for $w$, ALONE computes an element-wise product of $o$ and a filter vector $m_w$, and then applies a feed-forward network to increase its expressiveness. The two components are described as follows.

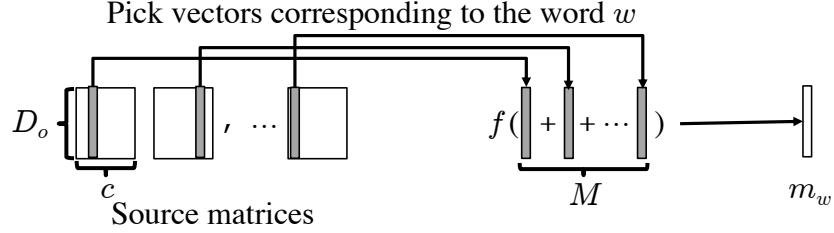

Pick vectors corresponding to the word $w$

Source matrices

Figure 2: Construction of the filter vector $m_w$. We pick several vectors corresponding to the word $w$ from source matrices and combine them to construct $m_w$.

## 2.1 Filter Construction

To make the result of the element-wise product $m_w \odot o$ unique to each word, we have to prepare different filter vectors from each other. In the simplest way, we sample values from a distribution such as Gaussian $D_o \times V$ times, and then we construct the matrix whose size is $D_o \times V$ with the sampled values. However, the above method requires similar memory space to $E$ in the conventional word embeddings. Thus, we introduce a more memory-efficient way herein.

Our filter construction approach does not prepare the filter vector for each word explicitly. Instead, we construct a filter vector by combining multiple vectors, as shown in Figure 2. In the first step, we prepare $M$ source matrices such as codebooks, each of which is $D_o \times c$. Then, we assign one column vector of each matrix to each word randomly. Thus, each word is tied to $M$ (column) vectors. In this step, since the probability of collision between two combinations is much small $(1 - \exp(-\frac{V^2}{2(c^M)})$ based on the birthday problem), each word is probably assigned to the unique set of $M$ vectors. Moreover, the required memory space, i.e., $D_o \times c \times M$ is smaller than $E$ when we use $c \times M \ll V$.

To construct the filter vector $m_w$, we pick assigned column vectors from each matrix, and compute the sum of the vectors. Then, we apply a function $f(\cdot)$ to the result. Formally, let $m_w^1, ..., m_w^M$ be column vectors assigned to $w$, we compute the following equation:

$$m_w = f\left( \sum_i^M m_w^i \right). \tag{1}$$

In this paper, we use two types of filter vectors: a binary mask and a real number vector based on the following distribution and function.

**Binary Mask**   A binary mask is a binary vector whose elements are 0 or 1, such as the dropout mask [29]. Thus, we ignore some elements of $o$ based on the binary mask for each word. For the binary mask construction, we use the Bernoulli distribution. To make the binary mask containing 0 with probability $p_o$, we construct the source matrices with sampling from the $\mathrm{Bernoulli}(1 - p_o^{\frac{1}{M}})$.

Moreover, we use the following $\mathrm{Clip}(\cdot)^2$ as the function $f$ to trim 1 or more to 1 in the filter vectors.

$$\mathrm{Clip}(a) = \begin{cases} 1 & 1 \le a \\ 0 & \text{otherwise} \end{cases} \tag{2}$$

In other words, in binary mask construction, $m_w$ is computed from the element-wise logical OR operation over all $m_w^i$.

**Real Number Vector**   In addition to the binary mask, we use the filter vector, which consists of real numbers. We use the Gaussian distribution to construct source matrices, and the identity transformation as the function $f$. In other words, we use the sum of vectors from source matrices without any transformation as the filter vector.

Table 1: Memory space required by each method for word representation. ALONE (naive) represents the case where we prepare filter vectors explicitly.

| Method | Memory usage |
|---|---|
| Conventional way | $D_e \times V$ |
| ALONE (naive) | $D_o + D_{inter} \times (D_o + D_e) + D_o \times V$ |
| ALONE (proposed) | $D_o + D_{inter} \times (D_o + D_e) + M \times D_o \times c$ |
| ALONE (proposed (volatile)) | $D_o + D_{inter} \times (D_o + D_e)$ |

## 2.2 Feed-Forward Network

We obtain the unique vector to each word by computing the element-wise product of $o$ and $m_w$. However, in this situation, words share the same value in several elements. Thus, we increase the expressiveness by applying a feed-forward network $\text{FFN}(\cdot)$ to the result of $m_w \odot o$:

$$\text{FFN}(x) = W_2(\max(0, W_1 x)), \tag{3}$$

where $W_1 \in \mathbb{R}^{D_{inter} \times D_o}$ and $W_2 \in \mathbb{R}^{D_e \times D_{inter}}$ are weight matrices. In short, the feed-forward network in this paper consists of two linear transformations with a ReLU activation function. We use the output of $\text{FFN}(m_w \odot o)$ as the word embedding for $w$.

## 2.3 Discussion on the Number of Parameters and Memory Footprint

Table 1 summarizes the memory space required by each method. In this table, ALONE (naive) ignores the filter construction approach introduced in Section 2.1. As described previously, the conventional way requires a large amount of memory space because $V$ is exceptionally large ($> 10^4$) in most cases. In contrast, ALONE (proposed) drastically reduces the number of parameters due to $D_o, D_{inter}, M \times c \ll V$, and thus, we can reduce the memory footprint when we adopt the introduced filter construction way. Moreover, since ALONE uses random initialized vectors as filter vectors without any training, we can reconstruct them again and again if we store a random seed. Thus, we can ignore the memory space for filter vectors such as ALONE (proposed (volatile)).

As an example, consider word embeddings on WMT 2014 English-to-German translation in the setting of the original Transformer [36]. In this setting, the conventional way requires 19M as the memory footprint due to $V = 37000$ and $D_e = 512$. In contrast, ALONE compresses the parameter size to 4M, which is less than a quarter of that footprint, when we set $D_o = 512$ and $D_{inter} = 4096$. These values are used in the following experiment on machine translation. For filter vectors, the naive filter construction way requires an additional 19M for the memory footprint, but our introduced approach requires only 262k more when we set $c = 64$ and $M = 8$. These values are also used in our experiments. Thus, the proposed ALONE reduces the memory footprint as compared to that of the conventional word embeddings.

# 3 Experiments

In this section, we investigate whether the proposed method, ALONE, can be an alternation of the conventional word embeddings. We first conduct an experiment on the reconstruction of pre-trained word embeddings to investigate whether ALONE is capable of mimicking the conventional word embeddings. Then, we conduct experiments on real applications: machine translation and summarization. We train the Transformer, which is the current state-of-the-art neural encoder-decoder model, combined with the ALONE in an end-to-end manner.

## 3.1 Word Embedding Reconstruction

In this experiment, we investigate whether ALONE has a similar expressiveness to the conventional word embeddings. We used the pre-trained 300 dimensional GloVe[3] [22] as source word embeddings and reconstructed them with ALONE.

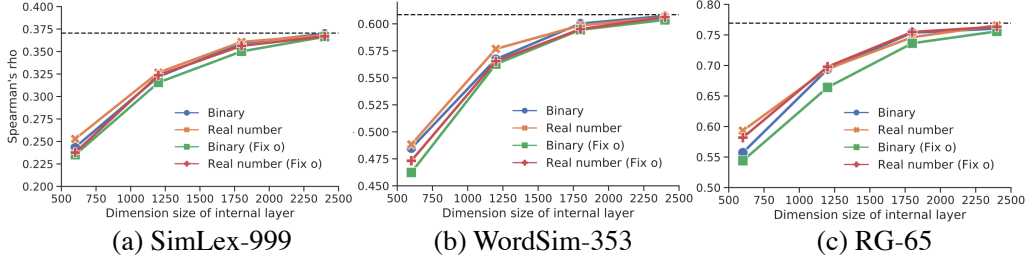

(a) SimLex-999  (b) WordSim-353  (c) RG-65

Figure 3: Results on word embedding reconstruction. The dashed line indicates the Spearman's rank correlation coefficient of GloVe.

**Training details:** The training objective is to mimic the GloVe embeddings with ALONE. Let $e_w$ be the conventional word embedding (GloVe in this experiment), we minimize the following objective function:

$$\frac{1}{V}\sum_{w=1}^{V}||e_w - \text{FFN}(m_w \odot o)||^2. \tag{4}$$

We optimized the above objective function with Adam [13] whose hyper-parameters are default settings in PyTorch [21]. We set mini-batch size 256 and the number of epochs 1000. We constructed each mini-batch with uniformly sampling from vocabulary and regard training on the whole vocabulary as one epoch. For $c$, $M$, and $p_o$ in the binary mask, we set 64, 8, and 0.5 respectively[4]. We used the same dimension size as GloVe (300) for $D_o$ and conducted experiments with varying $D_{inter}$ in $\{600, 1200, 1800, 2400\}$. In each setting, the total number of parameters is 0.4M, 0.7M, 1.1M, and 1.4M. We selected the top 5k words based on the frequency in English Wikipedia as target words[5]. We used five random seeds to initialize ALONE, and report the average of five scores.

**Test data:** We used the word similarity task, which is widely used to evaluate the quality of word embeddings [22, 4, 37]. The task investigates whether similarity based on trained word embeddings corresponds to the human-annotated similarity. In this paper, we used three test sets: SimLex-999 [9], WordSim-353 [5], and RG-65 [25]. We computed Spearman's rank correlation ($\rho$) between the cosine similarities of word embeddings and the human annotations as in previous studies.

**Results:** Figure 3 shows the Spearman's $\rho$ of ALONE in both filter vectors. This figure also shows the Spearman's $\rho$ without training $o$ (Fix $o$). The dashed line indicates the Spearman's $\rho$ of GloVe, i.e., the upper bound of word embedding reconstruction. Figure 3 indicates that ALONE achieved comparable scores to GloVe on all datasets in $D_{inter} = 2400$. These results indicate that ALONE has the expressive power to represent the conventional word embeddings.

In the comparison between filter vectors, we cannot find the significant difference in $D_{inter} = 2400$, but the real number vectors slightly outperformed binary masks in small $D_{inter}$. Thus, we should use real number vectors as the filter vectors if the number of parameters is strongly restricted.

Figure 3 shows that the setting without training $o$ is defeated against training $o$ in most cases. Therefore, it is better to train $o$ to obtain superior representations.

### 3.2 Machine Translation

Section 3.1 indicates ALONE can mimic pre-trained word embeddings, but there remain two questions:

1. Can we train ALONE in an end-to-end manner?
2. In the realistic situation, can ALONE reduce the number of parameters related to word embeddings while maintaining performance?

Table 2: Results of WMT En-De translation.

| Method | $D_{inter}$ | Embed | BLEU |
|---|---|---|---|
| ConvS2S [6] | | 66.0M | 25.2 |
| Transformer [36] | | 16.8M | 27.3 |
| Transformer+DeFINE [16] | | - | 27.01 |
| Transformer (conventional word embeddings) | | 16.8M | 27.12 |
| Transformer (factorized embed) | | 4.3M | 26.43 |
| Transformer (factorized embed) | | 8.5M | 26.56 |
| Transformer+ALONE (Binary) | 4096 | 4.2M | 26.97 |
| Transformer+ALONE (Real number) | 4096 | 4.2M | 26.93 |
| Transformer+ALONE (Binary) | 8192 | 8.4M | **27.55** |
| Transformer+ALONE (Real number) | 8192 | 8.4M | **27.61** |
| ALONE without training $o$ | | | |
| Transformer+ALONE (Binary) | 4096 | 4.2M | 26.75 |
| Transformer+ALONE (Real number) | 4096 | 4.2M | 26.85 |
| Transformer+ALONE (Binary) | 8192 | 8.4M | 26.90 |
| Transformer+ALONE (Real number) | 8192 | 8.4M | 26.95 |

To answer these questions, we conduct experiments on machine translation and summarization.

**Training details:** In machine translation, we used ALONE as word embeddings instead of the conventional embedding matrix $E$ in the Transformer [36]. We adopted the base model setting and thus shared embeddings with the pre-softmax linear transformation matrix. We used the fairseq implementation [19] and followed the training procedure described in its documentation [6]. We set $D_o$ the same number as the dimension of each layer in the Transformer ($d_{model}$, i.e., 512) and varied $D_{inter}$. For other hyper-parameters, we set as follows: $c = 64$, $M = 8$, and $p_o = 0.5$. Moreover, we applied the dropout after the ReLU activation function in Equation (3).

**Dataset:** We used WMT En-De dataset since it is widely used to evaluate the performance of machine translation [6, 36, 18]. Following previous studies [36, 18], we used WMT 2016 training data, which contains 4.5M sentence pairs, newstest2013, newstest2014 for training, validation, and test respectively. We constructed a vocabulary with the byte pair encoding [27] whose merge operation is 32K in sharing source and target. We measured case-sensitive BLEU with SacreBLEU [24].

**Result:** Table 2 shows BLEU scores of the Transformer with ALONE in the case $D_{inter} = 4096, 8192$. This table also shows BLEU scores of previous studies [6, 36, 16] and the Transformer trained in our environment with the same hyper-parameters as the original one [36][7]. DeFINE [16] uses a factorization approach, which constructs embeddings from small matrices instead of one large embedding matrix, to reduce the number of parameters[8]. In addition, we trained the Transformer with a simple factorization approach as a baseline. In the simple factorization approach, we construct word embeddings from one small embedding matrix and weight matrix to expand a small embedding to one with the larger dimensionality $D_e$. We modified the dimension size to make the number of parameters related to embeddings almost equal to those in ALONE.

Table 2 shows the Transformer with ALONE in $D_{inter} = 8192$ outperformed other methods even though the embedding parameter size is half that of the Transformer with the conventional embeddings. This result indicates that our ALONE can reduce the number of parameters related to embeddings without negatively affecting the performance on machine translation.

Table 3: Results on DUC 2004 task 1. The scores in bold are superior to the previous top score.

| Method | $D_{inter}$ | Embed | R-1 | R-2 | R-L |
|---|---|---|---|---|---|
| ABS [26] | | 42.1M | 28.18 | 8.49 | 23.81 |
| LSTM EncDec+WFE [32] | | 37.7M | 32.28 | 10.54 | 27.80 |
| Transformer+LRPE+PE [34] | | 8.3M | 32.29 | 11.49 | 28.03 |
| +ALONE (Binary) | 512 | 0.5M | 31.60 | 11.12 | 27.25 |
| +ALONE (Real number) | 512 | 0.5M | 31.96 | **11.50** | 27.74 |
| +ALONE (Binary) | 1024 | 1.0M | **32.51** | 11.48 | **28.08** |
| +ALONE (Real number) | 1024 | 1.0M | **32.57** | **11.63** | **28.24** |

In comparison with the factorized embedding approaches, Transformer with ALONE ($D_{inter} = 8192$) achieved superior BLEU scores as compared to Transformer+DeFINE [16], while the total parameter size of Transformer with ALONE, 52.5M, is smaller than it (68M). In other words, Transformer+ALONE achieved better performance even though it had a disadvantage in the parameter size. Moreover, Transformer+ALONE outperformed Transformer (factorized embed) despite the number of parameters related to embeddings in both being almost equal. These results imply that ALONE is superior to a simple approach using a small embedding matrix and expansion with a weight matrix.

Table 2 also shows BLEU scores of the Transformer with ALONE in the case without training the base embedding $o$. The setting without training $o$ didn't achieve BLEU scores as high as training $o$. Therefore, we also have to train $o$ in the training ALONE with an end-to-end manner.

### 3.3 Summarization

**Training details:** We also conduct an experiment on the headline generation task, which is one of the abstractive summarization tasks [26, 34]. The purpose of this task is to generate a headline of a given document with the desired length. Thus, we introduced ALONE into the Transformer with the length control method [34]. For other model details, we used the same as the experiment on machine translation. We used the publicly available implementation[9] and followed their training procedure. As the length control method, we used the combination of LRPE and PE [34]. Moreover, we re-ranked decoded candidates based on the content words following the previous study [34].

**Dataset:** As in previous studies [26, 34], we used pairs of the first sentence and headline extracted from the annotated English Gigaword with the same pre-processing script provided by Rush et al. [26][10] as the training data. The training set contains about 3.8M pairs. For the source-side, we used the byte pair encoding [27] to construct a vocabulary. We set the hyper-parameter to fit the vocabulary size 16K. For the target-side, we used a set of characters as a vocabulary to control the number of output characters. We shared the source-side and target-side vocabulary.

We used the DUC 2004 task 1 [20] as the test set. Following the evaluation protocol [20], we truncated the characters of the output headline to 75 bytes and computed the recall-based ROUGE score.

**Result:** Table 3 shows the ROUGE scores of the previous best method (Transformer+LRPE+PE [34]) with ALONE in the case $D_{inter} = 512, 1024$[11]. This table indicates that ALONE (Real number) achieved the comparable ROUGE-2 score to the previous best in $D_{inter} = 512$. In addition, Transformer+LRPE+PE+ALONE in $D_{inter} = 1024$ outperformed the previous best score except for ROUGE-2 in ALONE (Binary) despite the embedding parameter size being one-eighth of that of the original Transformer+LRPE+PE.

Figure 4 shows loss (negative log likelihood) values on validation sets of machine translation and summarization (newstest2013 and a set sampled from English Gigaword, respectively) for each word embedding method. This figure indicates that the convergence of the Transformer with ALONE is

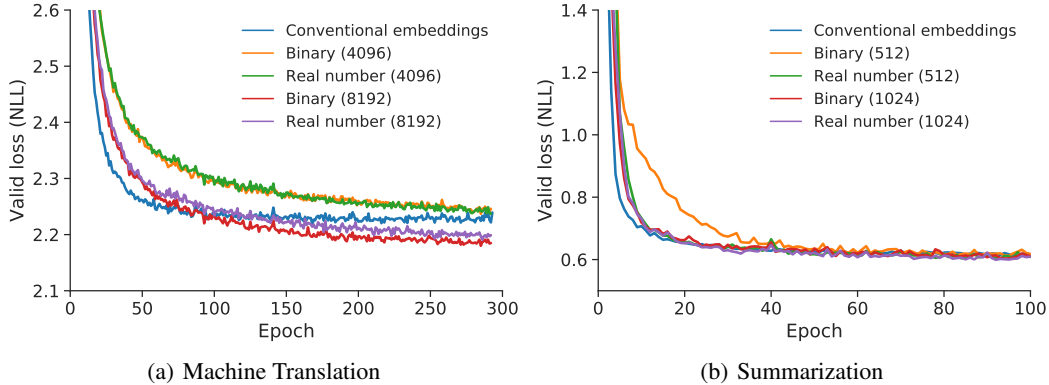

<div align="center">(a) Machine Translation      (b) Summarization</div>

<div align="center">Figure 4: Validation loss of each method.</div>

slower than the conventional embeddings. In particular, the convergence of Binary (512) is much slower than other methods in summarization. We consider this slow convergence harmed ROUGE scores of ALONE (Binary) in DUC 2004. Meanwhile, the Transformer+ALONE in $D_{inter} = 8192$, which outperformed the original Transformer in machine translation, achieved superior validation loss values as compared to the conventional embeddings.

In conclusion, based on the machine translation and summarization results, the answers for the previously mentioned two questions are as follows; 1. Yes, we can train ALONE with a neural method from scratch. 2. Yes, our ALONE can reduce the parameter size for embeddings without sacrificing performance.

# 4 Related Work

Researchers have proposed several strategies to compress neural networks such as pruning [15, 8, 39], knowledge distillation [10, 11], and quantization [7, 2, 28, 1, 31, 33]. Han et al. [8] proposed iterative pruning, which consists of the following three steps: (1) train a neural network to find important connections, (2) prune the unimportant connections, and (3) re-train the neural network to tune the remaining connections. Zhang et al. [39] iteratively performed steps (2) and (3) to compress a neural machine translation model.

Knowledge distillation approaches train a small network to mimic a pre-trained network by minimizing the difference between the outputs of the small network and pre-trained original network [10, 11]. Kim and Rush [11] applied knowledge distillation to a neural machine translation model and obtained a smaller model that achieves comparable scores to the original network.

These approaches require additional computational costs to acquire a compressed model because they need to train a base network before compression. In contrast, our proposed ALONE does not require a pre-trained network because we can train it with an end-to-end manner in the same way as the conventional word embeddings. In addition, we can also apply the above approaches to compress ALONE because the approaches are orthogonal to it.

Chen et al. [2] proposed HashedNet, which constructs the weight matrix from a few parameters. HashedNet decides the assignment of trainable parameters to an element of the weight matrix based on a hash function. Some studies applied such a parameter assignment approach to word embeddings [31, 28, 1]. For example, Suzuki and Nagata [31] constructs word embeddings with the concatenation of several sub-vectors (trainable parameters). Their method optimizes the parameters and assignments of sub-vectors through training. While those methods can represent each word with small parameters after training, they require additional parameters during the training phase. In fact, Suzuki and Nagata [31] uses conventional word embeddings during training. In contrast, such additional parameters are unnecessary in ALONE.

Svenstrup et al. [33] proposed the method to assign each word to several vectors using a hash function. Their method also has no additional parameters during training, but it requires weight vectors for each

word. Thus, its parameter size depends on the vocabulary size $V$. In contrast, since the parameters of ALONE are independent from $V$, ALONE can represent each word with fewer parameters.

To reduce the number of parameters related to word embeddings, some studies have reduced the vocabulary size $V$ [12, 37, 35]. In these days, it is common to construct the vocabulary with sub-words [27, 14] for generation tasks such as machine translation. Furthermore, some studies use characters (or character n-grams) as their vocabulary and construct word embeddings from character embeddings [12, 37, 35]. This study does not conflict with these studies because ALONE is used to represent the word, sub-word, and character embeddings in our experiments.

## 5   Conclusion

This paper proposes ALONE, a novel method to reduce the number of parameters related to word embeddings. ALONE constructs embeddings for each word from one embedding in contrast to the conventional way that prepares a large embedding matrix whose size depends on the vocabulary size $V$. Through experiments, we indicated that ALONE can represent each word while maintaining the similarity of pre-trained word embeddings. In addition, we can train ALONE in an end-to-end manner on real NLP applications. We combined ALONE with the strong neural encoder-decoder method, Transformer [36], and achieved comparable scores on WMT 2014 English-to-German translation and DUC 2004 very short summarization with fewer parameters.

## Broader Impact

This study addresses the reduction of trainable parameters for word embeddings. Word embeddings are fundamental component of various neural network-based NLP methods because we need them to convert a symbolic input into vector representations. Thus, the proposed method, ALONE, has potential to reduce the parameter size of existing neural network-based NLP methods. In this paper, we combined ALONE with a neural encoder-decoder model but we expect that it also has a positive effect on other methods such as large-scale neural language models.

## Acknowledgments and Disclosure of Funding

We thank Hiroshi Noji, Hitomi Yanaka, Koki Washio, and Saku Sugawara for constructive discussions. We thank anonymous reviewers for their useful suggestions. This work was supported by JSPS KAKENHI Grant Number JP18K18119. The first author is supported by Microsoft Research Asia (MSRA) Collaborative Research Program.

## Footnotes

[1]The code is publicly available at https://github.com/takase/alone_seq2seq

[2] Equation (2) represents the operation for a scalar value. If an input is a vector such as Equation (1), we apply Equation (2) to all elements in the input vector.

[3]https://nlp.stanford.edu/projects/glove/

[4]We tried several values, but we cannot observe any significant differences among the results.

[5]We can use the whole vocabulary of pre-trained GloVe embeddings but we restricted vocabulary size to shorten the training time.

[6]https://github.com/pytorch/fairseq/tree/master/examples/scaling_nmt

[7]The number of parameters for embeddings in the Transformer is different from that in the original one [36] owing to the difference of vocabulary size.

[8]Table 2 shows the reported score. We cannot demonstrate the embedding parameter size for Transformer+DeFINE because its vocabulary size is unreported but the parameter size of Transformer+DeFINE is 68M, which is larger than that of the original Transformer (60.9M). We consider that the original Transformer (and our experiments) save the total parameter size by sharing embeddings with the pre-softmax linear transformation matrix.

[9]https://github.com/takase/control-length

[10]https://github.com/facebookarchive/NAMAS

[11]In the abstractive summarization task, the vocabulary size is much smaller than that in the machine translation experiment because we used characters for the target-side vocabulary, and the source-side vocabulary size is also small. Thus, we set a smaller $D_{inter}$.

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
