[Reviews · NeurIPS 2020]

Review 1

Summary and Contributions: This paper presents a new method for performing neural network compression and decreasing the number of parameters required for creating word embeddings. The proposed method works by having a single base embedding for all words which then has a filter applied to it depending on the desired word for which to produce the embedding. The resulting vector is then passed through a simple neural network to produce the final word embedding. Much of the paper focuses on how to build these filters and evaluates the quality of the embedding along with its use in practical tasks.

Strengths: The evaluations provided provide strong evidence that this method for producing word embeddings using fewer parameters is effective. This work provides a novel (and effective) approach to parameter compression for neural networks. The NeurIPS community would likely be interested in and benefit from such a method, as it will feasibly allow for lower numbers of model parameters, thus reducing training time while maintaining model performance.

Weaknesses: In order to further demonstrate the benefit of this compression scheme in practice, it would have been nice to see results of training a larger network using the ALONE embeddings and comparing the performance with a standard transformer with a similar number of total parameters. Also, since one of the advantages of using a neural network compression scheme like this is the reduction in overall computation required for training an effective model, it would have been nice to see some comparisons in training time differences, computation time, total energy usage etc.

Correctness: Based on the descriptions in the paper, the methods used for evaluation of ALONE are correct and make use of standard evaluation datasets and setups.

Clarity: Overall, the paper is very clear and well written. There are some instances where ALONE is referred to as “the ALONE” or “our ALONE” which sounds a little odd.

Relation to Prior Work: A thorough comparison of this work with other approaches for reducing the number of parameters in neural networks is provided for both the experimental design as well as the theoretical contributions. The authors’ proposed method for compressing neural networks was compared with others such as pruning, knowledge distillation, and quantization. It was quite clear that ALONE took a novel approach to this issue.

Reproducibility: Yes

Additional Feedback: Why use this over pre-trained embeddings like GloVe, which are already expressive and fairly low-dimensional? UPDATE AFTER REBUTTAL: Thank you for addressing my concerns relative to GloVe. I stand by my score.


Review 2

Summary and Contributions: This paper proposes a method to compress the large word embedding matrix in NLP models to reduce the parameter size. The proposed approach computes word embeddings from shared filter vectors and another shared vector, making the parameter size independent from vocabulary size to achieve compression. Experiments on translation and summarization tasks demonstrate this method compresses embedding matrix significantly without losing performance. ------------ After Rebuttal ------------- Thank you for the response! It addressed most of my concerns and I would like to increase my score to 6.

Strengths: (1) The proposed method is very simple, it randomly assigns and combines filter vectors from a shared codebook and does not need to learn this discrete assignment operation (2) The results are good compared to non-compressed baselines with significantly less embedding parameters and on-par performance

Weaknesses: (1) I am not an expert in this direction, but I think the experiment lacks other compression baselines -- the main experiments only compare with normal seq2seq baseline and the toy “factorized embed” method while there are many embedding matrix compression methods out there as cited by the author. This paper compares to none of them, the only included DeFINE method does not show Embed params and it seems it is not in a comparable setting as well (as footnote says DeFINE uses more parameters than original transformer). In the introduction the authors exclude some related work like [1] saying they need additional parameters to learn, but I don’t think this is a reasonable excuse to not compare with them. Those methods are independent from the vocabulary size and have good performance/compression rate as well. Also, don’t the parameters of the feed-forward network here count as “additional parameters” ? I think those related work are comparable and the authors should compare at least one of other competitive compression methods to show the advantage of the proposed method over others. (2) It would be nice to show the overall compression rate instead of only the embedding matrix. Sometimes the overall compression rate might be small even though the embedding params seem to be compressed a lot. [1] Raphael Shu and Hideki Nakayama. Compressing word embeddings via deep compositional code learning. ICLR 2018

Correctness: Yes

Clarity: Yes

Relation to Prior Work: Yes

Reproducibility: Yes

Additional Feedback:


Review 3

Summary and Contributions: This paper describes a new memory-efficient method to represent word embeddings as opposed to storing them as word embedding matrix which can be huge. The main idea is to construct a function (or as they say a filter) that takes in a single common vector and outputs a word embedding. Specifically, this filter is defined by M matrices with c columns each which are randomly sampled from a predefined distribution. To get the vector for a word, one samples one column from each matrix, adds them and passes the resultant vector through a feedforward network. The matrices are initialized and fixed whereas the feedforward network is trained with the downstream tasks. The authors present an initial proof of concept with trying to reconstruct GloVe vectors. In their main experiments, they show competitive performance on WMT en-de and abstractive summarization (headline generation) both based on transformer models. They compare against embedding tables and factored embedding tables. The presented method requires less memory than baselines as performs equally well (sometimes better) than the baseline methods presented. ------------------------------ I thank the authors for their rebuttal and stand by my score.

Strengths: 1. The presented model requires less memory than traditional word embedding methods theoretically for really large vocabularies, which is quite impressive. 2. Experiments validate the utility of this method on multiple tasks.

Weaknesses: Not a limitation per se, but it would have interesting to see experiments on larger vocabulary settings (as for example with word based language models) where BPE is not used, since it decreases vocabulary size by a lot anyway. Something minor: The total memory calculation should include a VxM term which indicates the columns chosen from the matrices for each word.

Correctness: Yes, I believe so. The method seems sound and the experiments are done on two standard seq2seq tasks and datasets. The results also seem very convincing and impressive.

Clarity: Yes, very clearly written.

Relation to Prior Work: Yes, prior work seems quite extensive.

Reproducibility: Yes

Additional Feedback: Some analysis on what the embeddings learned from this model look like compared to traditional embedding tables would be interesting to see (maybe using certain intrinsic evaluation measures) As I said experiments on language modeling tasks would be interesting to see. Additionally, I think this work could have applications in training word embeddings (with fasttext like methods) or cross-lingual word embedding methods.


Review 4

Summary and Contributions: This paper proposed a novel word embedding method, ALONE, that reduces the parameter space without harming the performances on both word-level tasks and sophisticated NLP tasks.

Strengths: This paper resolves the problem that the number of word embedding parameters scales linearly with the vocabulary size, which is not affordable given the large vocabulary size of current training corpus. The proposed method addresses this problem by decomposing the word embeddings into a few codebooks, which requires less parameter space and can scale logarithmically with the vocabulary size. While the proposed method effectively reduces the parameter space, the performances of downstream applications are not affected. Some of the experiments even show better performances for specific architectures,

Weaknesses: The writing of this paper needs to be revised. Please explain the meaning of the notation before using it (e.g., D_{inter}). The proposed method directly deals with the pretrained GloVe. Why not jointly train the parameters using the GloVe objective? Also it would be nice to include the studies on applying the proposed method on top of pretrained language models (e.g. BERT/GPT) and see the performance.

Correctness: I have several concerns regarding the current experiment setups: 1. For reconstruction task, noticed that the performance is comparable to glove when D_{inter}=2400, the number of parameter in this setting is can embed approximately 5k words in GloVe, does this mean that the parameters can potentially overfit to the top 5k words? 2. If that's the case, it's worth reporting reporting the fraction of words that are in the top 5k frequent in English wikipedia in the simplex-999, wordsim-353, and RG-65 datasets. 3. To avoid overfitting, I'm curious to see the performance if (4) is applied across the entire vocabulary (probably weighted by the word frequency makes more sense than average). 4. Since the transformer model itself already has a lot of parameters, I don't think the size of word embeddings is the bottleneck. Reducing the word embeddings seems to be marginal according to line 195-200.

Clarity: The writing of this paper needs to be revised. Apart from some grammatical error, please explain the meaning of the notation before using it (e.g., D_{inter}).

Relation to Prior Work: yes

Reproducibility: Yes

Additional Feedback:

[Author Response · NeurIPS 2020]

We thank the reviewers for their careful reading and their many useful comments. If we could, we would like to respond
to each comment but the regulation of the conference limits the amount of our response. Thus, we focus on main
concerns from reviewers and answer them.

**Additional comparisons:** We appreciate advices from reviewers **#1**, **#3**, and **#4** on additional comparisons to further
indicate the effectiveness of the proposed method, ALONE. We are also intrigued to conduct such comparisons (e.g.,
training time, embedding analysis, and so on).

**Reviewer #4** are wondering whether ALONE can be used for language models like BERT/GPT. BERT/GPT consists of
the Transformer architecture that we used in our experiments. We indicated that the combination of Transformer and
ALONE works well in widely used machine translation and summarization datasets. Those results imply that ALONE
can be introduced in BERT/GPT and reduce the parameters related to their embeddings without negatively affecting the
performance.

**Word embedding reconstruction experiment:** We consider that reviewers **#1** and **#4** have some concerns about the
experiment on word embedding reconstruction. The motivation of this experiment is to investigate whether ALONE
has a similar expressiveness to the conventional word embeddings before real applications as described in Section
3. As pointed out by reviewers, we can train ALONE on a raw corpus based on the objective function of GloVe (or
other objectives such as skip-gram) and use the whole vocabulary in mimicking but we selected pre-trained GloVe
embeddings of 5k words as a target of mimicking to shorten the training time. We believe that it is more important to
investigate whether ALONE can reduce the parameter size related to embeddings in the real applications (Sections 3.2
and 3.3) to indicate the usefulness of ALONE.

As described in Section 3, we trained ALONE with an end-to-end manner in the experiments on machine translation
and summarization. In other words, we didn't use pre-trained embeddings and trained ALONE with Transformer jointly
from random initialization in contrast to prior studies such as Shu and Nakayama [2018] in these experiments.

**Lack of other compression baselines: Reviewer #2** considers that we didn't compare existing methods to reduce the
parameter size related to embeddings but we compared DeFINE (Mehta et al. [2020]) and the factorized embedding
approach. As described in Section 3.2, the total parameter size of Transformer+DeFINE is larger than ours. In WMT
En-De dataset, it is easy to achieve better performance for a model that has a large amount of parameters because
Transformer (big) outperforms Transformer (base) (Vaswani et al. [2017]). Thus, we would like to emphasize that
Transformer+ALONE achieved better performance than Transformer+DeFINE although Transformer+ALONE had a
disadvantage in the parameter size. In addition, **Reviewer #2** pointed out that the factorized embedding approach is toy
but this approach is used in the recent major work, ALBERT (Lan et al. [2020]), to reduce the embedding parameter
size. Therefore, we compared ALONE with approaches in recent studies.

**Reviewer #2** required the comparison with Shu and Nakayama [2018]. Indeed, they conducted experiments on machine
translation but their approach needs multiple training steps and additional parameters when we introduce it into neural
encoder-decoder models because their approach compresses 'pre-trained' embeddings. In fact, the training of Shu
and Nakayama [2018] consists of 3 steps in experiments on machine translation in their paper (training NMT model,
compressing embeddings, and re-training NMT model). In contrast, ALONE (and other compared methods in our
experiments) can be trained with an end-to-end manner based on the objective functions of the application tasks. Thus,
it is difficult to conduct a fair comparison because the training paradigms of ours and theirs are different. In other words,
we can combine ALONE with theirs if we have a large amount of time to construct a model.

**Definition of additional parameters: Reviewer #2** might be confused about the definition of "additional parameters".
As described in the line 43, "additional parameters" is the parameters required only during the training phase. For
example, the approach of Shu and Nakayama [2018] learns the mapping between primitive embeddings and words,
and deletes the parameters related to the mapping after the training. Moreover, their method requires pre-trained
embeddings. We call these parameters "additional parameters". Thus, the parameters of FFN in ALONE are not
"additional parameters", and the parameter sizes in Tables 2 and 3 include them.

**Compression rate of the whole parameter size:** We agree with reviewers **#2** and **#4** that it is also important to report
the compression rate of the whole parameter size in neural encoder-decoder models. However, since this study addresses
reducing the number of parameters related to embeddings, we consider that it is the most important to report the
embedding parameter size. The previous studies such as Shu and Nakayama [2018] and Chen et al. [2018] also reported
the number of parameters related to embeddings only (the reported "total size" in Shu and Nakayama [2018] includes
embeddings only). In addition, since ALONE is independent from an encoder-decoder architecture, we can combine
ALONE with any existing approach to reduce the parameter sizes of neural encoder-decoders. For example, we can
reduce the parameter size with cross-layer parameter sharing used in ALBERT (Lan et al. [2020]) but the reduction is
orthogonal to the proposed method.

[Meta-Review · NeurIPS 2020]

The paper proposed a novel mechanism to generate word embedings by a function rather than store the whole embeddings for each word, which leads to a high compression of space needed for word embeddings. The idea is novel and can be used broadly. The effectiveness is verified by the experiments. Reviewers suggest to conduct experiments in larger network and more comparison with baselines to show the saving of computation time, energy, etc.